# Multiplexed micronutrient, inflammation, and malarial antigenemia assessment using a plasma fractionation device

Eleanor Brindle [1,2], Lorraine Lillis[2], Rebecca Barney[2], Pooja Bansil[2], Francisco Arredondo [3], Neal E. Craft[4], Eileen Murphy[2], David S. Boyle [2]*

**1** Center for Studies in Demography and Ecology, University of Washington, Seattle, Washington, United States of America, **2** PATH, Seattle, Washington, United States of America, **3** Dept of Medicine, Duke University, Durham, North Carolina, United States of America, **4** Craft Nutrition Consulting, Elm City, North Carolina, United States of America

* dboyle@path.org

## Abstract

Processing and storing blood samples for future analysis of biomarkers can be challenging in resource limited environments. The preparation of dried blood spots (DBS) from finger-stick collection of whole blood is a widely used and established method as DBS are biosafe, and allow simpler field processing, storage, and transport protocols than serum or plasma. Therefore, DBS are commonly used in population surveys to assess infectious disease and/ or micronutrient status. Recently, we reported that DBS can be used with the Q-plex™ Human Micronutrient 7-plex Array (MN 7-plex), a multiplexed immunoassay. This tool can simultaneously quantify seven protein biomarkers related to micronutrient deficiencies (iodine, iron and vitamin A), inflammation, and malarial antigenemia using plasma or serum. Serum ferritin, an iron biomarker, cannot be measured from DBS due to red blood cell (RBC) ferritin content confounding the results. In this study, we assess a simple blood fractionation tool that passively separates plasma from other blood components via diffusion through a membrane into a plasma collection disc (PCD). We evaluated the concordance of MN 7-plex analyte concentrations from matched panels of eighty-eight samples of PCD, DBS, and wet plasma prepared from anticoagulated venous whole blood. The results showed good correlations of >0.93 between the eluates from PCD and DBS for each analyte except ferritin; while correlations seen for plasma/PCD were weaker. However, the recovery rate of the analytes from the PCD were better than those from DBS. The serum ferritin measures from the PCD were highly correlated to wet plasma samples (0.85). This suggests that surveillance for iron status in low resource settings can be improved over the current methods restricted to only measuring sTfR in DBS. When used in combination with the MN 7-plex, all seven biomarkers can be simultaneously measured using eluates from the PCDs.

**Data Availability Statement:** All of the data generated in this study can be publicly accessed at Dataverse (https://doi.org/10.7910/DVN/VY3MDB).

**Funding:** DSB; OPP1154343; Bill & Melinda Gates Foundation; https://www.gatesfoundation.org/ NC; OPP1135483; Bill & Melinda Gates Foundation; https://www.gatesfoundation.org/ EB; P2C HD042828; Eunice Kennedy Shriver National Institute of Child Health and Human Development; https://www.nichd.nih.gov/ The funders had no role in study design, data collection and analysis, decision to publish, or preparation of the manuscript.

**Competing interests:** The authors have declared that no competing interests exist.

## Introduction

Micronutrient deficiency (MND) has been described as hidden hunger and an estimated 2 billion people are affected globally [1] with iodine, iron, and vitamin A representing three of the four primary MNDs of global significance [2–4]. Sufficient concentrations of key micronutrients (MN) are required for normal development of the fetus, infants, and young children. MND can impair cognitive, immune, and ocular function and physical health and development. Fetuses, young children, and women of reproductive age are particularly at risk from MND and the burden is greatest in low- and middle-income countries (LMICs) that are the least equipped to routinely assess MN status [5].

There are low cost and effective solutions to mitigate the effects of MND including dietary initiatives, fortification of common foodstuffs (e.g. salt or flour) or directly via supplementation [6–8]. The challenge to nutrition programs and researchers in LMICs is to effectively perform population surveillance to identify groups at most risk of MND before its effects manifest. Baseline data can identify the scale of MND and groups at greatest risk so that effective interventions can be introduced. Interventions can then be monitored via ongoing surveillance at scheduled intervals to establish their impact.

There is general consensus on a panel of blood-based biomarkers that are both informative on iodine, iron and vitamin A status [9–13], and practical for use in population-level surveillance. These act as surrogates to more direct MN measurement methods which involve challenging specimen collection (e.g. bone marrow aspirates for iron measurement [14]) or require highly skilled users and complex equipment (e.g. high pressure liquid chromatography for serum retinol analysis [15]). The blood specimens for these tests can be collected by venipuncture or less invasive methods, such as finger- or heel-stick capillary blood collection. Venous blood collection offers the advantage of much larger sample volumes and reduced chance of error resulting from the capillary blood collection technique (e.g. milking fingers), but venous samples are more difficult to collect, particularly from children, and more challenging to transport, process, and store [16]. Serum or plasma from whole blood must be rapidly processed to avoid hemolysis and must be kept under cold chain to prevent decay of the biomarkers. A perception of greater pain associated with venipuncture versus finger stick collection may also negatively influence participation in studies [17]. Blood and its derivatives pose biohazard risks, and handling should only be performed by staff with certified training for blood borne pathogens [18]. Therefore, the scaled collection of venous blood adds significant logistical, financial and technical challenges particularly in low resource settings where the need for effective MND surveillance is greatest.

The use of DBS as an alternative sample type meets many of the challenges presented with venous whole blood. Blood collection of up to 250 μL from a heel or finger prick via sterile lancet can be performed by less skilled users [19]. The blood is spotted on a paper card and left to dry before being stored in a sealed pouch with a desiccant to eliminate moisture. The preparation of a sample takes less time, needs no ancillary equipment (e.g. centrifuge or refrigeration) and uses fewer materials and consumables [16, 20, 21]. Once dried, the sample has greatly reduced biohazard risk during sample handling and processing. DBS cards have the further advantages of being relatively small, requiring significantly less space in freezers during long-term storage. As many biomarkers are stable in DBS at a broader range of temperatures, they can be shipped using gel packs instead of dry ice, or even out of cold chain [19, 20, 22–24].

A primary challenge in using DBS is that the sample quantity is limited, with a typical collection yielding five spots of whole blood with approximately 50 μL each. Eluting DBS from filter paper results in dilution of the specimen, so assays must be capable of quantifying low concentrations. We recently highlighted that low concentration biomarkers such as

thyroglobulin (Tg, μg/L) could be quantified from DBS eluates using the Q-Plex™ Human Micronutrient Array (Quansys Biosciences, Logan, Utah, USA; hereafter the MN 7-plex) [25, 26]. The MN 7-plex is a multiplexed immunoarray that enables the simultaneous quantification of biomarkers to deficiencies in iodine, iron, and vitamin A, in addition to inflammatory biomarkers and *Plasmodium falciparum*, from a single small volume of liquid serum or plasma [25].

While this evaluation found acceptable agreement between DBS and plasma for most analytes, using DBS for iron deficiency assessment is challenging because ferritin levels are grossly elevated by the co-elution of serum and red blood cell ferritin from lysed RBCs. Therefore, sTfR is the only iron biomarker that can be measured from DBS [26, 27]. While sTfR is one of the recommended biomarkers for iron status assessment, we and other groups have reported challenges in harmonizing absolute sTfR values derived from different immunoassay methods [2, 25–29]. The ratio of sTfR to serum ferritin has been found to be a better indicator of iron deficiency than either analyte alone [2]. As such, we hypothesize that a parallel measurement of ferritin, inflammatory biomarkers and sTfR may give greater confidence in establishing iron status. To support this, we propose that devices developed to passively fractionate plasma from whole blood in austere settings, primarily for viral load testing of people living with HIV [19, 22], could be repurposed to include serum ferritin measurement and each of the other assays included on the MN 7-plex immunoarray.

In this manuscript we describe an evaluation of use of the MN 7-plex with matched sets of plasma, eluates of DBS, and eluates from a prototype of the plasma collection disc (PCD), a passive blood fractionation device [24]. In particular, we evaluate whether the PCD allows reliable measurement of ferritin using a method with the same logistical advantages as DBS.

## Materials and methods

### Collection of whole blood specimens

Specimens of ~8 mLs of human whole blood were procured from a commercial vendor, BioIVT (Westbury, New York). The WIRB protocol number for the collection of 80 blood samples was 20161665 and all specimens were obtained only after signed consent by each participant. The PATH Office of Research Affairs also deemed the use of these samples to be non-human subjects research.

### Preparation of DBS, PCD and plasma panels

A panel of 80 heparinized anticoagulated whole blood samples (40 adult male, 40 adult female) was used for this study. The whole blood was delivered to the PATH laboratory the next day, and processed upon receipt to prepare three sample types (DBS, PCD, and plasma) within 24 hours of blood collection. Upon arrival in the laboratory, blood was continuously rotated at 4 ˚C before processing to ensure it remained thoroughly mixed. Matched sets of plasma, DBS and PCD were prepared in triplicate from the eighty blood specimens and the complete panels were measured in three different laboratories for α-acid glycoprotein (AGP), C-reactive protein (CRP), ferritin, histidine rich protein 2 (HRP2), retinol binding protein 4 (RBP4), soluble transferrin receptor (sTfR), and thyroglobulin (Tg) using the MN 7-plex. Because the blood panel was derived from US donors, it was not expected to include malarial parasitemia or a high prevalence of inflammation or iron deficiency [26]. Thus, twenty whole blood samples were randomly chosen from the set and a portion of the whole blood was spiked with calibrator solutions provided by Quansys to simulate samples positive for HRP2 and with elevated AGP, CRP, or sTfR levels. Each contrived sample was prepared by transferring 2 mL of whole blood to a new tube, adding spike solutions, and gently mixing by rotating tubes at 4 ˚C for at

**Table 1. Contrived blood samples spiked with a range of elevated levels of AGP, CRP, HRP2 and sTfR.**

| Spike | Analyte | Spike Level | # samples | Spiking solution added (µL) | Total Volume | Expected added concentration |
|-------|---------|-------------|-----------|------------------------------|--------------|------------------------------|
| 1 | HRP2 | Very High | 5 | 11.7 | 70 | 3.40 µg/L |
| | sTfR | High | | | | 160 mg/L |
| 2 | HRP2 | High | 5 | 3.9 | 30 | 0.843 µg/L |
| | sTfR | Mid | | | | 13 mg/L |
| 3 | HRP2 | Mid | 5 | 18.8 | 110 | 0.286 µg/L |
| | CRP | High | | | | 120 mg/L |
| 4 | HRP2 | Low | 5 | 144.8 | 750 | 0.045 µg/L |
| | AGP | High | | | | 2.0 g/L |

least 5 minutes (see Table 1). The resulting samples (Sample numbers 81–100) were then processed for DBS and PCD as described below (Fig 1).

DBS were prepared from the 80 whole blood specimens and 20 spiked whole blood specimens by spotting 70 µL of whole blood onto Whatman 903 cards (n = 100). The cards were stored overnight to dry at room temperature and then individually stored at -80°C in sealed plastic pouches with desiccant packets.

Prototype PCDs were assembled from parts supplied by the manufacturer for use in these experiments. To produce the PCD, each absorbent plasma collection disc was loaded onto a custom-made manifold and covered by a separation membrane (Fig 2). A volume of 35 µL of whole blood was added to each PCD membrane using a wide-bore pipet tip. After approximately 90 minutes, the PCD membranes containing the RBCs and other cellular components were removed and discarded. The PCDs containing the adsorbed plasma were then removed from the manifold and stored in zipped plastic bags with desiccant at -80 °C. Samples 63–80 were omitted from this portion of the testing for lack of PCD units; the resulting final panel included PCDs from 62 unadulterated donor specimens and 20 PCDs from spiked specimens (total n = 82). Significant hemolysis was noted on one PCD (ID 12), and corresponding DBS spots appeared very pale. Some slight hemolysis above background was observed on PCDs for 13 samples (Samples 4, 5, 32, 37, 38, 50, 52, 83, 85, 86, 94, 96, and 100). Remaining whole blood was then processed to collect plasma fractions (n = 100) as previously described [27]. These plasma fractions were aliquoted into cryovials and stored at -80°C until use. The PCDs were eluted at PATH. The eluted material from the triplicate samples was pooled and mixed,

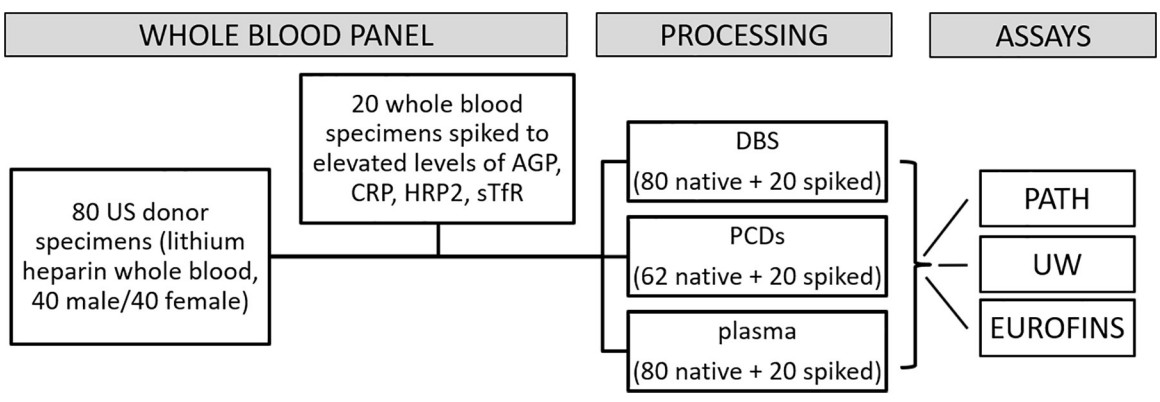

**Fig 1. Sample panels and experiment design.**

and then aliquots of approximately 180 μL were sent to each laboratory on dry ice (see next section), along with matched DBS cards and plasma samples.

## Preparation of eluates from DBS and PCDs

The DBS and PCD samples were eluted in the Q-plex assay sample diluent before undergoing analysis with the MN 7-plex assay. DBS processing was carried out in each laboratory the day before assay. Two 6 mm DBS punches, equivalent to approximately 6 μL of serum [23], were placed in 244 μL sample diluent (approx. 1:20 dilution) and left overnight at 4°C. Samples were then shaken at 200 rpm on a microplate shaker for one hour at room temperature before using. PCD processing occurred in one laboratory, with aliquots of elutes then shared with all 3 laboratories. Each PCD containing an estimated equivalent of approximately 8 to 10 μL plasma per PCD was placed in a microcentrifuge tube separation column with a paper filter membrane (Pierce Spin Cups, ThermoFisher Scientific, Waltham, MA, USA) and 200 μL of diluent was added to each PCD, resulting in a dilution of approximately 1 in 20. Samples were incubated at room temperature for two hours before being centrifuged at 12,500 rpm for 15 minutes. Three PCDs per sample were eluted following this procedure, and recovered eluates were pooled for each sample, mixed and aliquoted in ~180 μL volumes for transfer to the laboratories for testing.

## Human micronutrient 7-plex array procedure

The MN 7-plex protocol instructs the user to prepare all samples, negative control and calibrator dilutions in sample diluent containing reconstituted competitor. Because the competitor may degrade during the incubation time required for DBS or PCD elution, the standard protocol was modified to add competitor to the mix after sample elution. These protocol alterations were also used for wet plasma specimens included in these analyses.

Competitor was reconstituted in one tenth of the volume of diluent suggested in the MN 7-plex manual to produce a 10X competitor stock and 144 μL of eluate from either DBS or PCD was mixed with 16 μL of 10X assay competitor mix resulting in 1X competitor in a final sample dilution of approximately 1 in 22. Similarly, kit diluent without competitor was used to dilute wet plasma specimens to 1 in 20, and to reconstitute the calibrator provided with the kit and prepare the standard curve; 10X competitor mix was added to prepared solutions as 10% of the final volume. Then, 50 μL of each solution was added into the test wells of the MN 7-plex assay in duplicate, and the remainder of the protocol was carried out as directed in the kit manual.

After addition of the standards and samples, each plate was incubated at room temperature for two hours with shaking at 500 RPM using a flatbed shaker (Titertek Berthold, Huntsville, Alabama, USA). All reactions were aspirated and washed 3 times with 400 μL of wash buffer supplied with the MN 7-plex kit using an automatic plate washer. Next, 50 μL of detection mix was added to each well and the plate was then incubated with shaking at 500 RPM for 1 hour and washed one more time as described above. 50 μL streptavidin horseradish peroxidase solution was added to each well and plates were incubated with shaking for 20 minutes. Following another wash as described above, the chemiluminescent substrate parts A and B were mixed in equal volumes and 50 μL of the mixture was then added to each well.

Each plate was then imaged at 270 seconds of exposure time using a Quansys Q-View™ Imager LS (Quansys Biosciences). Q-View™ Software (Quansys Biosciences) was used to overlay a plate map onto the locations of analyte spots in each well and to measure the chemiluminescent signal from each spot in units of pixel intensity. The software applies the calibrator concentration values to the pixel intensities for each spot in the standard curve wells and fits 5

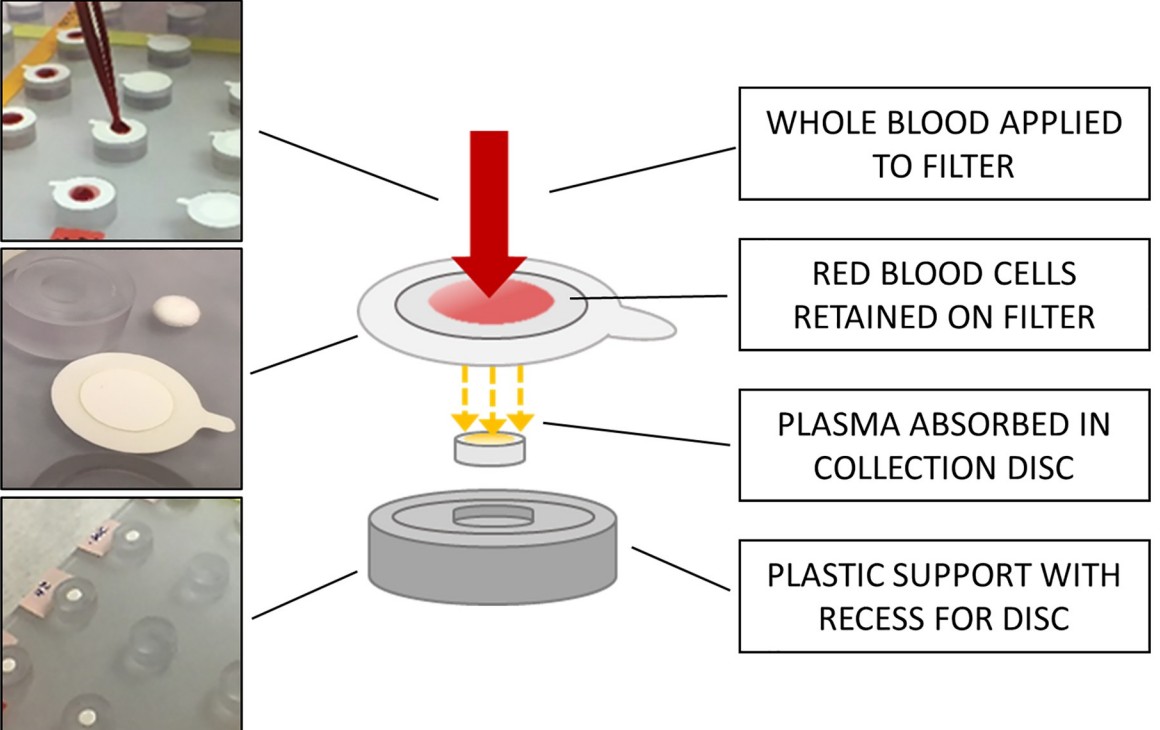

**Fig 2. Schematic illustrating the plasma collection disc in operation.** The schematic illustrates a single PCD. Photos on the left show the PCD with the the filter disc placed on a manifold for multiple samples and the addition of 35 μL of whole blood to each disc (top), a PDC detail view prior to whole blood being added (middle), and the discs after blood has been absorbed and finally the plasma collection disc exposed after the whole blood treated filter has been removed (bottom). The disc is removed with sterile forceps and is ready for use or can be stored until required.

parameter logistic calibration curves for each analyte. The pixel intensities of the spots in each test well are then used to interpolate the concentration of each analyte relative to its calibrator curve. Once the plate image is overlaid with the analysis grid, all of the curve fitting and data reduction steps are automatically applied via the software. The upper and lower limits of quantification determined by Quansys for each kit lot were applied to exclude values beyond the assay concentration ranges. All values were adjusted for dilutions.

## Adjusting volume estimates for the PCD

The original concentration estimates for the PCD eluates used the assumption that each disc contained approximately 10 μL of plasma, resulting in a dilution of 1 in 22 when the disc was eluted. However, recovery results suggested that the volume of plasma recovered from each disc is approximately 50% of the original estimate, giving an eluate dilution of 1 in 44. To account for this, the PCD results were doubled to more accurately reflect the eluted plasma volume.

## Statistical methods

Reproducibility across the three laboratories was determined using coefficient of variation (CV, standard deviation divided by the mean) of the three measures for each sample type. The CVs were also used to assess the inter-assay and inter-laboratory variation. Concordance between AGP, CRP, ferritin, HRP2, RBP4, sTfR, and Tg results in plasma with both DBS (100

pairs) and PCD (82pairs) was evaluated using scatter plots, Lin's concordance correlation coefficient (CCC) and Spearman's correlation coefficient [30]. Agreement in absolute value was evaluated by calculating recovery of plasma values from DBS or PCD (DBS or PCD concentration/plasma concentration, expressed as a percentage) and using Bland Altman plots of the average concentration plotted against the percentage difference in concentrations of each analyte in plasma/DBS or PCD sample pair. Statistical analyses were conducted using Stata 15.1 (StataCorp, College Station, TX, USA). All of the data generated in this study can be publicly accessed at Dataverse (https://doi.org/10.7910/DVN/VY3MDB).

## Results

Summary statistics for each analyte and sample type and CVs for the measures across the three participating laboratories are shown in Table 2. The CVs for plasma and DBS were similar, with higher CVs in the PCDs for most analytes. The Eurofins sTfR values were substantially greater than the other datasets and after review, this data was omitted from further analysis. The CVs were also high for HRP2, regardless of sample type, because of the semi-quantitative nature of the assay. Most specimens in the panel contained no detectable HRP2; spiked samples contained high values near or above the upper limit of detection for the assay. For ferritin, the DBS samples saturated the assay and were at the upper limit of detection, leading to high

**Table 2. Summary statistics and cross-laboratory coefficients of variation (CV) for plasma, DBS and PCD.**

| | | Plasma | | | DBS | | | PCD | | |
|---|---|---|---|---|---|---|---|---|---|---|
| | | N | Mean (SD) | Average CV | N | Mean (SD) | Average CV | N | Mean (SD) | Average CV |
| AGP (g/L) | All | 100 | 1.1 (0.6) | 10.9% | 100 | 1.2 (0.6) | 10.8% | 82 | 0.6 (0.2) | 8.7% |
| | Unspiked | 80 | 1.0 (0.2) | 11.5% | 80 | 1.1 (0.3) | 11.3% | 62 | 0.5 (0.2) | 7.0% |
| | Spiked | 20 | 1.55 (1.2) | 8.7% | 20 | 1.7 (1.1) | 9.0% | 20 | 0.7 (0.4) | 14.1% |
| CRP (mg/L) | All | 92 | 7.44 (12.3) | 19.1% | 89 | 5.5 (8.6) | 16.0% | 73 | 2.8 (4.1) | 18.3% |
| | Unspiked | 73 | 4.93 (5.4) | 20.6% | 70 | 3.6 (4.1) | 18.5% | 54 | 1.9 (2.0) | 20.5% |
| | Spiked | 19 | 17.08 (22.9) | 13.4% | 19 | 12.2 (15.3) | 7.2% | 19 | 5.3 (7.0) | 13.2% |
| Ferritin (µg/L) | All | 99 | 42.9 (40.6) | 17.7% | 100 | 178.8 (114.1) | 11.6% | 75 | 18.4 (16.0) | 23.6% |
| | Unspiked | 79 | 43.6 (43.5) | 15.6% | 80 | 178.9 (115.2) | 12.0% | 56 | 19.0 (17.3) | 26.5% |
| | Spiked | 20 | 39.9 (27.2) | 25.8% | 20 | 178.3 (112.4) | 10.2% | 19 | 16.7 (11.3) | 16.1% |
| HRP2 (µg/L) (semi-quantitative) | All | 26 | 3.59 (4.8) | 18.6% | 100 | 0.79 (2.1) | 60.7% | 49 | 0.9 (1.5) | 52.2% |
| | Unspiked | 6 | 0.17 (0.1) | 12.2% | 80 | 0.10 (0.0) | 62.2% | 29 | 0.2 (0.0) | 92.6% |
| | Spiked | 20 | 4.61 (5.1) | 19.0% | 20 | 3.55 (3.5) | 59.4% | 20 | 1.9 (2.0) | 45.8% |
| RBP4 (µmol/L) | All | 100 | 1.67 (0.6) | 12.2% | 100 | 1.20 (0.3) | 9.2% | 82 | 0.6 (0.2) | 10.3% |
| | Unspiked | 80 | 1.65 (0.6) | 12.7% | 80 | 1.16 (0.3) | 10.4% | 62 | 0.6 (0.2) | 9.5% |
| | Spiked | 20 | 1.74 (0.6) | 10.5% | 20 | 1.34 (0.4) | 4.4% | 20 | 0.7 (0.2) | 12.6% |
| sTfR (mg/L)* | All | 100 | 48.6 (41.3) | 35.9% | 100 | 82.5 (42.0) | 9.9% | 81 | 35.2 (19.8) | 34.3% |
| | Unspiked | 80 | 44.7 (39.7) | 34.1% | 80 | 80.8 (43.8) | 10.5% | 61 | 33.6 (20.1) | 34.6% |
| | Spiked | 20 | 64.0 (45.1) | 43.0% | 20 | 89.0 (34.3) | 7.4% | 20 | 39.9 (18.7) | 33.2% |
| Tg (µg/L) | All | 98 | 23.8 (22.1) | 12.2% | 99 | 21.1 (21.7) | 8.7% | 81 | 12.4 (13.3) | 27.5% |
| | Unspiked | 78 | 22.6 (20.5) | 13.1% | 79 | 20.0 (19.6) | 9.2% | 61 | 11.2 (10.7) | 28.0% |
| | Spiked | 20 | 28.4 (27.7) | 8.9% | 20 | 25.5 (28.9) | 6.7% | 20 | 16.1 (19.2) | 26.2% |

Spiked samples contain added AGP, CRP, HRP2, and/or sTfR as shown in Table 1. The concentrations are mean values from testing at PATH, UW, and Eurofins.

*Eurofins data are excluded from sTfR results.

Average CV is the average of the individual sample CVs calculated from results across the 3 laboratories. Abbreviations: AGP, α-1-acid glycoprotein; CRP, C-reactive protein; CV, coefficient of variation; DBS, dried blood spots; N, number; PCD, plasma collection disc; RBP4, retinol-binding protein 4; SD, standard deviation; sTfR, soluble transferrin receptor; Tg, thyroglobulin.

CVs. This was expected given the presence of red blood cell (RBC) associated ferritin present in this sample type. Hemolysis was a concern based on the assumption that these samples may contain RBC ferritin. However, the 14 PCD samples with observed hemolysis after the blood filtration step did not show any evidence of effects on the ferritin concentrations. Concentrations of all samples, spiked, and unspiked, were similar for plasma and DBS, but the PCD values were approximately half the values measured in plasma when dilution was calculated using the original estimate of plasma volume contained in the PCD. Results were adjusted twofold to account for the new plasma volume equivalent estimate. This gave improved agreement with plasma values in terms of absolute values but had minimal impacts on correlation (Fig 3A and 3B).

The Spearman correlations between plasma and DBS and between plasma and PCDs are shown in Table 3. The Spearman correlation coefficients ($R_s$) between the plasma and the DBS samples were high (0.93 or higher), with the exception of ferritin. These results closely match previous observations using DBS samples with the 7-plex assay [22]. As expected, the correlation between plasma ferritin and DBS ferritin was lower (0.64) as DBS eluates contain a mixture of ferritin from serum and from red blood cells. The correlation of plasma ferritin to PCD ferritin was 0.925, highlighting the potential advantage of PCDs for ferritin measurement without the need to centrifuge samples and possibly for dry plasma samples. The correlation for RBP4 from PCD was lower than from DBS (0.993 versus 0.840). The sTfR correlation using the PATH and UW data only data was 0.858.

To complete our comparative analysis, we also assessed the correlation of the DBS and the adjusted PCD samples to the plasma samples using scatter plots with linear regression lines (Fig 3A and 3B) and their agreement using Bland Altman plots (Fig 4A and 4B). We used Lin's CCC to create scatter plots to compare the measurements [31]. AGP and Tg were both very close to the line of perfect concordance and sTfR is correlative though the reduced major axis is consistently greater than the line of perfect correlation. HRP2, CRP, and RBP4 are less concordant while bias increases showing greater variance at the higher analyte concentrations. The plot for ferritin shows the lowest correlation as is expected since DBS samples have ferritin contributed from RBCs. With the exception of the sTfR plot, all other DBS analytes tended to be lower than plasma values. For most analytes, the correlation of adjusted PCD and plasma was similar to that of DBS and plasma.

## Discussion

Serum ferritin is a good biomarker for the immunologic determination of iron status from blood. However, RBC ferritin can confound analysis via quantitative ELISA, so blood samples must be fractionated into serum or plasma soon after collection, which is labor intensive and challenging in austere settings. The primary goal of this study was to assess the potential for the non-centrifugal preparation of plasma using a passive blood fractionation tool for its subsequent measurement in a multiplexed immunoassay than can assess MN status, inflammation and malaria antigenemia. We successfully demonstrated that a prototype passive PCD tool could prepare samples of sufficient quality for serum ferritin measurement and that were highly concordant via $R_s$ (0.925) using paired plasma samples.

Over 18% of the PCDs also had visible signs of hemolysis after processing but these did not noticeably affect the ferritin test data. We did observe a clear trend of lower-than-expected recovery values for all of the biomarkers within the PCD dataset. This discrepancy in volume likely arose because while a PCD disc can absorb 10 µL of plasma if directly added to it, the PCD cannot absorb the same volume of plasma via passive diffusion when wicking plasma from the whole blood sample. This was resolved after we adjusted the dilution factor to reflect

(A)

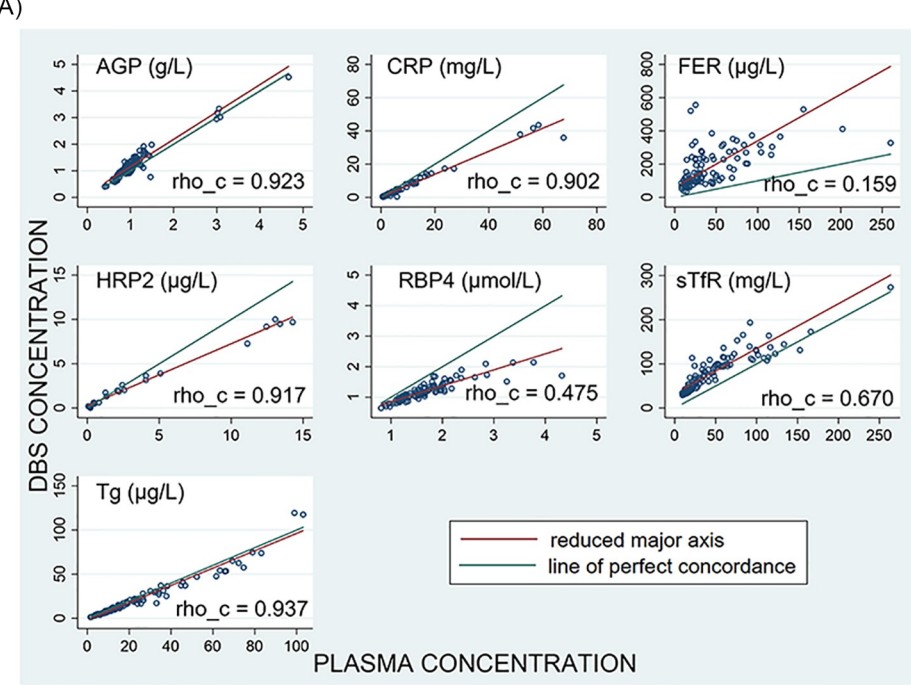

(B)

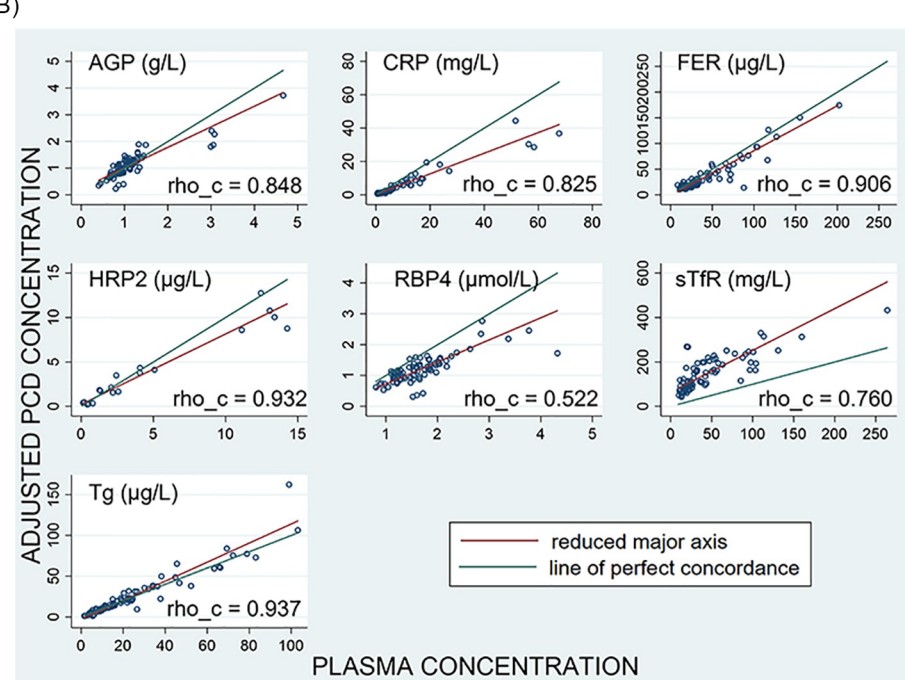

**Fig 3. A**. Lin's concordance correlation coefficient (CCC) plots comparing the analyte measurements in paired wet plasma (x-axes) and DBS samples (y-axes). AGP, α-1-acid glycoprotein; CRP, C-reactive protein; FER, ferritin; HRP2, histidine rich protein 2; RBP4, retinol binding protein 4; sTfR, soluble transferrin receptor; Tg, thyroglobulin; DBS, dried blood spot; rho_c, CCC. **B**, Lin's CCC plots comparing the analyte measurements in paired wet plasma (x-axes) and the adjusted PCD (y-axes). AGP, α-1-acid glycoprotein; CRP, C-reactive protein; FER, ferritin; HRP2, histidine rich protein 2; RBP4, retinol binding protein 4; sTfR, soluble transferrin receptor; Tg, thyroglobulin; DBS, dried blood spot; rho_c, CCC.

**Table 3. The comparison of results for plasma, DBS and the volume adjusted PCD eluates using $R_s$ and recovery of expected plasma values.**

|  | AGP | CRP | Ferritin | HRP2 | RBP4 | sTfR* | Tg |
|---|---|---|---|---|---|---|---|
| DBS |  |  |  |  |  |  |  |
| N (pairs) | 100 | 89 | 99 | 26 | 100 | 100 | 98 |
| $R_s$ | 0.975 | 0.988 | 0.640 | 0.977 | 0.993 | 0.947 | 0.995 |
| Recovery, mean (SD%) | 116 (15) | 71 (14) | 570 (400) | 98 (55) | 74 (9) | 215 (76) | 89 (11) |
| PCD |  |  |  |  |  |  |  |
| N (pairs) | 82 | 71 | 75 | 21 | 82 | 81 | 81 |
| $R_s$ | 0.835 | 0.965 | 0.848 | 0.914 | 0.760 | 0.855 | 0.964 |
| Recovery mean (SD%) | 101 (21) | 79 (36) | 90 (29) | 103 (66) | 73 (16) | 176 (69) | 102 (24) |

*Eurofins data are excluded from sTfR results.

** $R_s$ p-value <0.0001 for all comparisons.

Abbreviations: AGP, α-1-acid glycoprotein; CRP, C-reactive protein; DBS, dried blood spots; N, number; PCD plasma collection disc; RBP4, retinol-binding protein 4; $R_s$, Spearman's correlation coefficient; SD, standard deviation; sTfR, soluble transferrin receptor; Tg, thyroglobulin; Rho, rank-order correlation.

an assumption that the plasma volume contained in a PCD is approximately 5 μL, and not 10 μL as originally estimated. Further recovery experiments are warranted to confirm the new PCD plasma volume estimate.

When comparing plasma with DBS samples, ferritin failed as expected, but the $R_s$ for the other biomakers were 0.947 or higher. For recovery values, the DBS results were variable with AGP, HRP, and Tg being within <16% of the pooled plasma values. Slight under-recovery was seen with CRP (71%) and RBP4 (74%) and lastly, sTfR demonstrated significant over-recovery at 215%. Therefore, with the exception of ferritin and sTfR, the DBS data matches our previously reported observations [25, 26]. When comparing the PCD data with plasma, the $R_s$ values for the PCD and plasma were not as correlative as seen with DBS with exception of ferritin (0.84). The poorest PCD correlation was for RBP4 at 0.76 whilst the DBS samples gave a near perfect $R_s$ of 0.993. The best correlations from PCD derived analytes were for CRP and Tg at 0.96. It is unsurprising that the correlation between the plasma and DBS results were nearly identical as both have accurate volumes of plasma or blood applied. With the PCDs, the $R_s$ were more discordant than for DBS because while precise volumes of blood were applied to the PCD filters, the exact volumes of plasma retained in the PCDs were unknown. HRP2 was recovered well from both DBS and the PCD eluates, suggesting that the malarial antigen assay is functional with either DBS or PCD samples. For recovery, the PCD data was generally improved versus the DBS after a correction factor was applied. One reason may be that the plasma stored in the PCD can readily diffuse into the buffer whereas with the DBS there is some natural attrition of the biomarkers during drying, retention of analytes in the fibrous dead space within the paper, and some irreversible binding of proteins to the nitrocellulose matrix. While our results are preliminary, they suggest use of PCDs may allow ferritin measurement in plasma filtrates, while also yielding results for other biomarkers similar to those from plasma generated by centrifugation.

It is evident from the mean recoveries that PCDs either contained ~50% less plasma or that a significant amount of material was retained within the disc. Therefore, a rigorous assessment of the volumetric capacity of the PCD when wicking from whole blood is warranted, essentially mimicking the efforts to assess the volume of material retained in DBS relative to sampling volumes [21]. For convenience, the specimens used in this study were derived from a healthy USA-based cohort and so are unlikley to have micronutient deficencies and have no exposure to *P. falciparum* malaria. Therefore future assessments with more representative populations

(A)

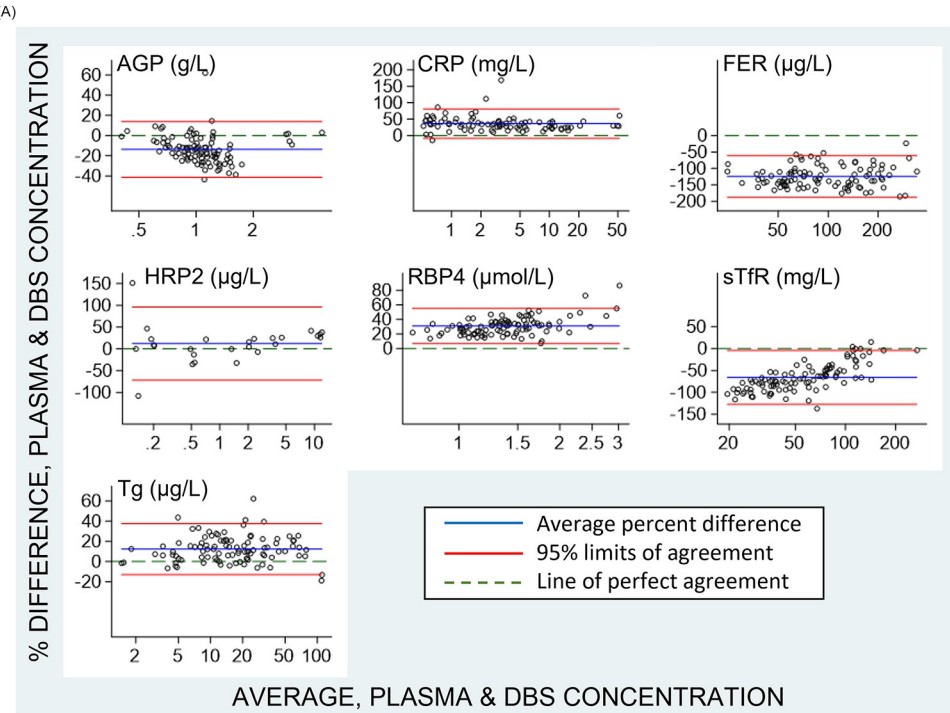

(B)

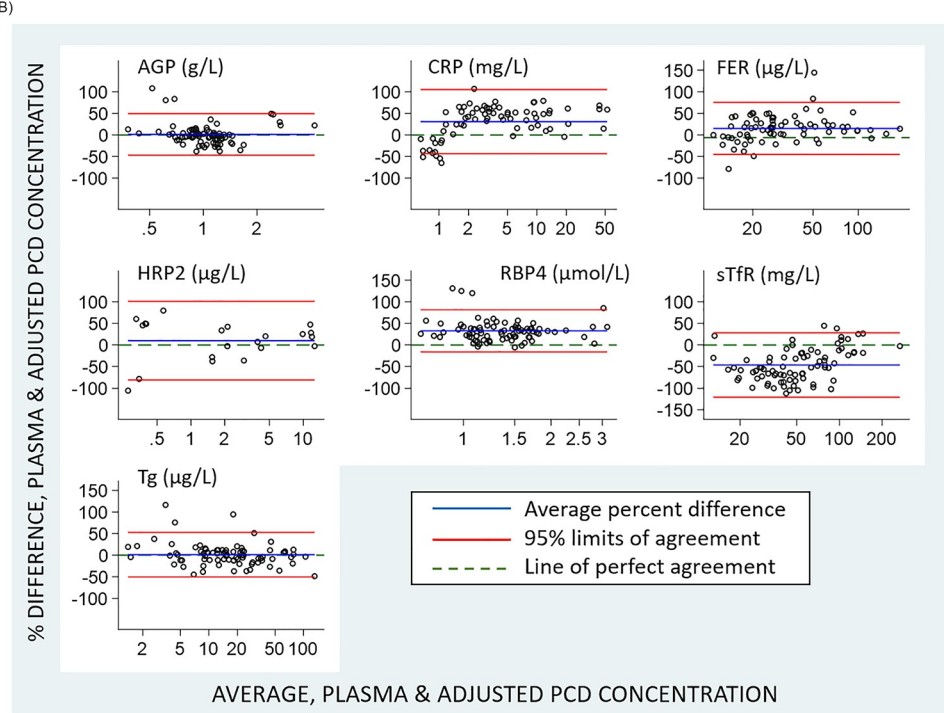

**Fig 4. A**. Bland Altman plots of the difference in results derived from wet plasma versus DBS types measured using the 7-plex assay. Average of the wet plasma and dry sample type (x-axes) are plotted against the difference between measurements from the 7-Plex assay for paired wet plasma and dry sample types (y-axes). **B**. Bland Altman plots of the difference in results derived from wet plasma versus the adjusted PCD. For both Figures, the horizontal lines indicate line of perfect agreement (green), mean (purple), and ± 2standard deviations of the difference (red). AGP, α-1-acid glycoprotein; CRP, C-reactive protein; FER, ferritin; HRP2, histidine rich protein 2; RBP4, retinol binding protein 4; sTfR, soluble transferrin receptor; Tg, thyroglobulin; DBS, dried blood spot.

from LMICs are necessary. While we added experimental rigor by performing the duplicate testing at three sites, we could not reconcile the sTfR values from being grossly elevated from one group wherein all of the other biomarkers remained in range. In general, the sTfR assay in the MN-7-plex array has performed the most poorly with plasma, DBS and PCD sample types [25, 26, 32]. We reported the poor performance of the sTfR immunoassay pair to the manufacturer, who now claim to have successfully improved performance with a new sTfR antibody pair. We are currently assessing the performance of the previous and new version of the MN 7-plex array to independently confirm this.

Our results highlight that there is generally good agreement between the sample types and that the measurement of seven of the analytes in the MN 7-plex array is feasible with a sample collection method that eliminates the need for equipment-based sample processing in the field and that may potentially minimize cold chain requirements with storage as a dried plasma collection card. Future efforts will be required to understand the stability of the biomarkers in the dried PCDs in their intended use for population-based micronutrient surveys in austere settings where access to refrigeration is absent. We aim to validate the performance of these tools in LMIC settings to better understand their performance and limitations. A passive blood fractionation tool to enable HIV viral load testing from dried plasma spots is commercially available (Cobas Plasma Separation card, Roche Diagnostics, Indianapolis, IN, USA) [33]. It remains to be seen if these can be succesfully incorporated with immunoassays such as the MN 7-plex but it highlights that other tools may exist to support effective population-based surveillance of micronutrient status in austere settings.

## Acknowledgments

We greatly appreciate the collaborative support of ViveBio LLC (Alpharetta, GA, USA) in supplying prototype PCD devices and for their expert technical input. We would like to thank Dr Emily Smith (The George Washington University, DC, USA) and Dr Ken Brown (University of California Davis, CA, USA) for their critical thinking and advice. We would also like to thank Ms. Olivia Halas for her administrative support in preparing and reviewing the manuscript for submission.

## Author Contributions

**Conceptualization:** Eleanor Brindle, Neal E. Craft, David S. Boyle.

**Data curation:** Eleanor Brindle, Pooja Bansil.

**Formal analysis:** Eleanor Brindle, Pooja Bansil, Neal E. Craft.

**Funding acquisition:** Eleanor Brindle, Neal E. Craft, David S. Boyle.

**Investigation:** Eleanor Brindle, Lorraine Lillis, Rebecca Barney, Pooja Bansil, Francisco Arredondo.

**Methodology:** Eleanor Brindle, Lorraine Lillis, Rebecca Barney, Francisco Arredondo, Neal E. Craft, David S. Boyle.

**Project administration:** Eileen Murphy.

**Supervision:** Eleanor Brindle, Neal E. Craft, Eileen Murphy, David S. Boyle.

**Writing – original draft:** Eleanor Brindle, David S. Boyle.

**Writing – review & editing:** Eleanor Brindle, Lorraine Lillis, Rebecca Barney, Pooja Bansil, Francisco Arredondo, Neal E. Craft, Eileen Murphy, David S. Boyle.

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
