## [Decision Letter · Decision Letter 0]

30 Jun 2022

PONE-D-21-34846Multiplexed micronutrient, inflammation, and malarial antigenemia assessment using a plasma fractionation devicePLOS ONE

Dear Dr.Boyle

Thank you for submitting your manuscript to PLOS ONE. After careful consideration, we feel that it has merit but does not fully meet PLOS ONE’s publication criteria as it currently stands. Therefore, we invite you to submit a revised version of the manuscript that addresses the points raised during the review process.

ACADEMIC EDITOR: Please follow the suggestion of the review. There are several point highlighted in the revision of the manuscript that needs to be resolved by authors for further consideration by PLOS ONE'sPlease submit your revised manuscript by July, 29

Please include the following items when submitting your revised manuscript:A rebuttal letter that responds to each point raised by the academic editor and reviewer(s). You should upload this letter as a separate file labeled 'Response to Reviewers'.A marked-up copy of your manuscript that highlights changes made to the original version. You should upload this as a separate file labeled 'Revised Manuscript with Track Changes'.An unmarked version of your revised paper without tracked changes. You should upload this as a separate file labeled 'Manuscript'.

We look forward to receiving your revised manuscript.

Kind regards,

José Luiz Fernandes Vieira

Academic Editor

PLOS ONE

Journal Requirements:

Reviewers' comments:

Reviewer's Responses to Questions

**Comments to the Author**

1. Is the manuscript technically sound, and do the data support the conclusions?

Reviewer #1: Partly

2. Has the statistical analysis been performed appropriately and rigorously? 

Reviewer #1: I Don't Know

3. Have the authors made all data underlying the findings in their manuscript fully available?

Reviewer #1: Yes

4. Is the manuscript presented in an intelligible fashion and written in standard English?

Reviewer #1: Yes

5. Review Comments to the Author

Reviewer #1: This study addresses a practical question that is of particular relevance to those studying micronutrient deficiencies in the global context. The primary question is whether a plasma collection disk (PCD) that separates whole blood into plasma at the time of sample collection offers the ability to measure ferritin, in particular, in addition to other multiplexed analytes, as well as wet plasma. Being able to use the PCD devices would offer the possibility of collecting and shipping whole blood samples without having to undergo steps that require a centrifuge and cold chain, and would be an improvement of dried blood spots of whole blood, which cannot be used for ferritin assessment. The study that was conducted was relatively straightforward, and a considerable amount of detail is shared, making it somewhat challenging to synthesize major findings. Some points for consideration are:

1) In the abstract, would it be possible to share more quantitative information? For example, in line 32, what makes for “high correlation”.

2) The Introduction is somewhat long and detailed, but information is logically presented. Some minor editorial issues are:

a. Line 52, “iron and vitamin A” instead of “iron or vitamin A”

b. Line 70, “spotted on a paper card”

c. Line 72, “fewer” instead of “less”

d. Lines 74-77, run-on sentence that is hard to read. Would divide into 2 sentences at line 75 after “storage”.

e. Lines 101-103, was the intent of the study to use heparinized plasma or was the use of heparin just a consequence of how the plasma was obtained? Leaving the introduction on this note implies that a comparison of EDTA versus heparin is a goal of the study, but nothing to this point in the Intro has led the reader to the conclusion that this comparison would be necessary, and it doesn’t come up again, even in the Discussion.

3) In the Methods at Lines 146-148, it seems that information about the quality of the samples (hemolysis) could be construed as a result, particularly given that 15 PCDs were affected by some degree of hemolysis that could have implications for the analyte measurements (particularly ferritin). Were these hemolyzed samples considered when the results were presented comparing the PCD to wet plasma?

4) Line 153/Figure 2, Figure 2 is very difficult to interpret. It would make more sense to make it supplemental and/or replace with some type of conceptual schematic to demonstrate how the product is intended to work. The figure itself does not help my understanding of the method.

5) There are a considerable number of comparisons to consider in the Results. Should the comparison of ferritin by DBS versus PCD and wet plasma methods (Lines 243-245) even be reported since the use of DBS for ferritin was dismissed in the Introduction? If TfR in the Eurofins lab was so different than in the UW and PATH labs could it just be eliminated from the results with an explanation that inter-lab variability was deemed too high and thus only data from two labs were considered to provide the best case scenario for that analyte (Lines 232-233; 239-240; 265-266)? Could explaining in the methods the iterative process of estimating the volume eluted from the PCD that led to presenting the “adjusted” PCD results rather than explaining this process throughout the Results section (Lines 260-276) and showing both approaches in Table 3? It seems like streamlining some of what the reader needs to wade through to get to the essence of the comparisons of interest would be helpful.

6) Lines 276 and 289-290 in Results and 317-318 in the Discussion have a strange error message (Error! Reference source not found.) in my version of the paper that needs to be resolved.

7) In Figures 3 and 4, the PCD approach is referred to as VIVEBIO, which is different than how it is named elsewhere throughout the paper, including the Figure legends.

8) For Figure 3, it would be very helpful to have correlation coefficients and p-values on the figures themselves. The correlation coefficients are in Table 3, but I don’t see p-values anywhere.

9) For Figure 4, it might be useful to express variability across analytes in common units like % difference relative to concentration.

10) Lines 288-291 in Results describe methods.

11) Lines 298 to end of Results. Text at line 298 reads like a figure legend—ie, it is an incomplete sentence. Perhaps, although it refers to Figure 3A, it was actually intended to refer to Figure 4A? And the text at line 301 is intended to be the figure legend for Figure 4B? If so, there is no text interpretation of those figures, and the headings at lines 306 and 309 make no sense. This needs to be cleaned up.

12) The Discussion seemingly starts with a page-long paragraph (Lines 313-333) that does not provide a general overview before diving into and repeating results, including referring back to Tables and Figures. Given all the comparisons done in the paper, it would be useful to re-orient the reader to the original intent and highlight a) lessons learned regarding the PCD technology (hemolysis? challenges in assessing volume represented, etc), b) an analyte-by-analyte description of findings with respect to plasma (as gold standard) vs PCD vs DBS, including particular inter-lab challenges of the sTfR assay c) study strengths (highly controlled, multi-lab, forward thinking) and limitations (population sampled might not represent the populations for whom the assays most needed), d) final recommendations.

13) The information in Lines 348-353 about a new sTfR assay coming out seems to undermine presenting sTfR in this paper at all.

14) In Line 367, the paper is ended on the note that all analytes can be measured “reliably” using the PCD with the MN 7-plex approach. How do you define “reliably”? The data do not seem to support that assertion—eg, Table 2. Perhaps saying that there is generally good agreement, in reference to the correlations, would be a more accurate way to interpret the findings.

6. PLOS authors have the option to publish the peer review history of their article (what does this mean?). If published, this will include your full peer review and any attached files.

Reviewer #1: No

---

## [Author Response · Author response to Decision Letter 0]

10 Sep 2022

Responses to the reviewer’s comments

Firstly, the project team would like to thank the reviewer of this manuscript most sincerely for taking the time and effort to review our draft and to add some excellent comments highlighting weaknesses within it. We greatly appreciate the provision of some excellent thoughts around how we can strengthen the quality of our work for the reader to better interpret and enjoy. As a result, we have endeavoured to follow their advice where possible.

1) In the abstract, would it be possible to share more quantitative information? For example, in line 32, what makes for “high correlation”.

Response: We have made some extensive edits to the Abstract in order to create some word space to then change the text to reflect the reviewer’s thinking where we can add some datapoints to be supportive of the generalized descriptions of the data. We have briefly reported on the two primary measurements made for correlation to plasma samples and also the recovery of the biomarkers as compared to the plasma data.

2) The Introduction is somewhat long and detailed, but information is logically presented. Some minor editorial issues are:

a. Line 57, “iron and vitamin A” instead of “iron or vitamin A”. We made the recommended change.

b. Line 75, “spotted on a paper card” We made the recommended change.

c. Line 77, “fewer” instead of “less” We made the recommended change.

d. Lines 79-82, run-on sentence that is hard to read. Would divide into 2 sentences at line 75 after “storage”. We made the recommended change, see Line 80.

e. Lines 101-103, was the intent of the study to use heparinized plasma or was the use of heparin just a consequence of how the plasma was obtained? Leaving the introduction on this note implies that a comparison of EDTA versus heparin is a goal of the study, but nothing to this point in the Intro has led the reader to the conclusion that this comparison would be necessary, and it doesn’t come up again, even in the Discussion.

Response: We thank the reviewer for pointing out this discrepancy. We have looked at our existing data and we unable to make a direct comparison at this time and so we have deleted this sentence 

3) In the Methods at Lines 146-148, it seems that information about the quality of the samples (hemolysis) could be construed as a result, particularly given that 15 PCDs were affected by some degree of hemolysis that could have implications for the analyte measurements (particularly ferritin). Were these hemolyzed samples considered when the results were presented comparing the PCD to wet plasma?

Response: This is a quite excellent point raised by the reviewer and we are very pleased to have an opportunity to amend our writing. Firstly, after review of our laboratory notebooks, the number of hemolyzed samples was 14 and not 15 as first reported. We have amended the text on Line 148 to inform the reader as to the ID# of the remaining 13 samples so they can review the raw and analyzed datasets themselves. We also now clearly note that the ferritin assay data was unaffected by the hemolysis from these as compared to the ferritin assay being off scale due to gross hemolysis from dried blood spots. See Line 243 – 245

“However, the hemolysis observed in 14 samples after the blood filtration step did not noticeably affect the measurement of ferritin concentration, a concern based on extra RBC ferritin possibly being in these samples.”

4) Line 153/Figure 2, Figure 2 is very difficult to interpret. It would make more sense to make it supplemental and/or replace with some type of conceptual schematic to demonstrate how the product is intended to work. The figure itself does not help my understanding of the method.

Response: We have followed the reviewer’s recommendation and have now added a schematic alongside the original photographs to clearly illustrate the component parts of the product and how it is intended to work. Some of the text in the figure title has also been amended to better inform the reader. See Lines 159 -165. 

5) a. There are a considerable number of comparisons to consider in the Results. Should the comparison of ferritin by DBS versus PCD and wet plasma methods (Lines 243-245) even be reported since the use of DBS for ferritin was dismissed in the Introduction?

Response: We appreciate the point that the reviewer is making in that this is not new data. However, we would like to retain this data as it highlights the excellent performance of the PCD in preparing a very acceptable sample for ferritin measurement versus using a dry blood spot from the same blood sample. The primary point of our study. 

5) b. If sTfR in the Eurofins lab was so different than in the UW and PATH labs, could it just be eliminated from the results with an explanation that inter-lab variability was deemed too high and thus only data from two labs were considered to provide the best case scenario for that analyte (Lines 232-233; 239-240; 265-266)?

Response: We appreciate this particularly good point to save the reader from superfluous text and so we will leave the full data set including the EuroFins sTfR data in the Dataverse file. In the relevant text in Tables 2 and 3 we now focus only on the data that is fit for analysis and comment. We have amended Lines 251-255 in the header for Table 2 and Lines 260-266 for Table 3 in both cases noting the sTfR data excludes Eurofins. We have added a sentence on Lines 235-237 noting the data discrepancy and the dropping of the Eurofins set. 

The Eurofins sTfR values were significantly greater than the other datasets and after review, this data was omitted from further analysis.”

The next sentence, Lines 236 -238, was deleted as it is now redundant for explaining the Eurofins data and the preceding sentence noted all CVs were high. We have also deleted the sentence on Line 269 that noted the sTfR data variation between labs and made changes in Lines 275 – 278.

“The correlation for RBP4 from PCD was lower than from DBS (0.993 versus 0.840). The sTfR correlation using the PATH and UW data only data was reasonable (0.858).”

5) c. Could explaining in the methods the iterative process of estimating the volume eluted from the PCD that led to presenting the “adjusted” PCD results rather than explaining this process throughout the Results section (Lines 260-276) and showing both approaches in Table 3? It seems like streamlining some of what the reader needs to wade through to get to the essence of the comparisons of interest would be helpful.

Response: We have accepted the reviewer’s recommendation to simplify the text and so have created a section in the M&M wherein we note that the PSC values were doubled in order to compensate for a 50% loss in volume as compared to the wet samples and DBS eluates. This is represented in Lines 213 – 218 and Lines 243 – 249. In moving this text, we also moved the Figures 3A and 3B to after Table 2 to accommodate the text noting that the reader should read the figures. See Line 260 - 266.

6) Lines 276 and 289-290 in Results and 317-318 in the Discussion have a strange error message (Error! Reference source not found.) in my version of the paper that needs to be resolved.

Response: These were error messages associated with the RefMan reference manager that we used. Each of these error messages have been resolved after we replaced the RefMan system with Zotero throughout the entire manuscript. We did not use track changes to record this as it would have created a very challenging document to read and/or further edit.

7) In Figures 3 and 4, the PCD approach is referred to as VIVEBIO, which is different than how it is named elsewhere throughout the paper, including the Figure legends.

Response: We have revised Figures 3B and 4B to use PCD and so make this consistent through the entire manuscript and its associated documents.

8) For Figure 3, it would be very helpful to have correlation coefficients and p-values on the figures themselves. The correlation coefficients are in Table 3, but I don’t see p-values anywhere.

Response: We agree with these suggestions, and Lin’s Rho is now shown on the concordance correlation coefficient plots. The Spearman correlation coefficients shown in Table 3 now include a footnote indicating that the p-value is <0.0001 for all comparisons. See line 284.

9) For Figure 4, it might be useful to express variability across analytes in common units like % difference relative to concentration.

Response: We have revised the Bland-Altman plots to use percent difference on the y-axis. The scales for all the y-axes are also adjusted to center each plot on zero to make it easier to compare percent difference across analytes and sample types.

10) Lines 288-291 in Results describe methods.

Response: We have relocated this text from Line 293 – 298 to Lines 213 – 218 in the Methods section.

11) Lines 298 to end of Results. Text at line 298 reads like a figure legend—i.e., it is an incomplete sentence. Perhaps, although it refers to Figure 3A, it was actually intended to refer to Figure 4A? And the text at line 301 is intended to be the figure legend for Figure 4B? If so, there is no text interpretation of those figures, and the headings at lines 306 and 309 make no sense. This needs to be cleaned up.

Response: We fully agree with the reviewer’s comments both here and earlier in this same body of text. Due to multiple edits and the relocation of text to other areas of the Results or Methods sections, the entire section from Lines 289 - - 314 is now deleted.

12) The Discussion seemingly starts with a page-long paragraph (Lines 313-333) that does not provide a general overview before diving into and repeating results, including referring back to Tables and Figures. Given all the comparisons done in the paper, it would be useful to re-orient the reader to the original intent and highlight a) lessons learned regarding the PCD technology (hemolysis? challenges in assessing volume represented, etc.), b) an analyte-by-analyte description of findings with respect to plasma (as gold standard) vs PCD vs DBS, including particular inter-lab challenges of the sTfR assay c) study strengths (highly controlled, multi-lab, forward thinking) and limitations (population sampled might not represent the populations for whom the assays most needed), d) final recommendations.

Response: We have rewritten the entire discussion including the reviewer’s recommendations to better describe our goals, results, conclusions, and limitations of our work. This can be found from Line 402 to Line 468.

13) The information in Lines 348-353 about a new sTfR assay coming out seems to undermine presenting sTfR in this paper at all.

Response: This was a recommendation by the project team to Quansys after this (and other ) poor correlative data for sTfR and so the work we present was our motivator for change. We present the three other references to sTfR being the least accurate of the 7 assays on the array and these results were the motivation for us to really push the developer to completely redesign sTfR as they were previously reluctant to invest in this. We have amended the text, now on Lines 449- 453 to reflect this. In reality, while the assay was redeveloped, the complete validation testing took place after our manuscript was submitted. The newer version of the 7 plex Human micronutrient assay including the sTfR modification was released onto the market by Quansys on June 1st, 2022. PATH are currently preparing a manuscript that highlights the improved performance sTfR, in addition to other assays including ferritin, CRP and RBP4 via direct comparison to the V1.0 kits.

“In general, the sTfR assay in the MN-7-plex array has performed the most poorly with plasma, DBS and PCD sample types [25,26,32]. We reported the poor performance of the sTfR immunoassay pair to the manufacturer who now claim to have successfully improved performance with a new sTfR antibody pair. We are currently assessing the performance of the previous and new version of the MN 7-plex array to independently confirm this.”

One of the other papers we reference here was under review for publication during submission of this work and so we did not refer to it in this manuscript. However, in that work sTfR was also the poorest performing analyte on the array and we have added it below for the reviewer.

Brindle E, Lillis L, Barney R, Bansil, P et al., Wessells KR, Ouedraogo CT, et al. A multicenter analytical performance evaluation of a multiplexed immunoarray for the simultaneous measurement of biomarkers of micronutrient deficiency, inflammation and malarial antigenemia. PLoS One. 2021; Nov 4;16(11):e0259509.

14) In Line 367, the paper is ended on the note that all analytes can be measured “reliably” using the PCD with the MN 7-plex approach. How do you define “reliably”? The data do not seem to support that assertion—e.g., Table 2. Perhaps saying that there is generally good agreement, in reference to the correlations, .would be a more accurate way to interpret the findings.

Response: The reviewer is correct. We have amended this text in the rewrite of the discussion and the reviewed sentence can be found at Lines 457 – 460.

“Our results highlight that there is generally good agreement between the sample types and that the measurement of seven of the analytes in the MN 7-plex array is feasible with a sample collection method that eliminates the need for equipment-based sample processing in the field and potentially minimize cold chain requirements.”

---

## [Decision Letter · Decision Letter 1]

14 Oct 2022

PONE-D-21-34846R1Multiplexed micronutrient, inflammation, and malarial antigenemia assessment using a plasma fractionation devicePLOS ONE

Dear Dr. David S. Boyle

Thank you for submitting your manuscript to PLOS ONE. After careful consideration, we feel that it has merit but does not fully meet PLOS ONE’s publication criteria as it currently stands. Therefore, we invite you to submit a revised version of the manuscript that addresses the few points raised during the review process. A few points of reviewer should be adressed for publication.

ACADEMIC EDITOR:

We look forward to receiving your revised manuscript.

Kind regards,

José Luiz Fernandes Vieira

Academic Editor

PLOS ONE

Journal Requirements:

**Comments to the Author**

1. If the authors have adequately addressed your comments raised in a previous round of review and you feel that this manuscript is now acceptable for publication, you may indicate that here to bypass the “Comments to the Author” section, enter your conflict of interest statement in the “Confidential to Editor” section, and submit your "Accept" recommendation.

Reviewer #1: (No Response)

2. Is the manuscript technically sound, and do the data support the conclusions?

Reviewer #1: Yes

3. Has the statistical analysis been performed appropriately and rigorously? 

Reviewer #1: Yes

4. Have the authors made all data underlying the findings in their manuscript fully available?

Reviewer #1: Yes

5. Is the manuscript presented in an intelligible fashion and written in standard English?

Reviewer #1: Yes

6. Review Comments to the Author

Reviewer #1: General: I am glad the authors found comments to be useful, and the responses and updated manuscript demonstrates that comments were thoughtfully considered. I did not see a track-changed version of the paper--it seemed like the original paper with updated figures and then a clean version of the updated paper were appended in the pdf I received--so comments below are based on what seemed to be the clean copy of the revision. Also, some of the issues that were pointed out in my original comments remain, so I hope that it was not a penultimate version that was inadvertently uploaded that I reviewed.

Minor points—

Line 45: delete “in”

Lines 97-99: sentence starting with “A variety of simple devices….” seems out of the blue; perhaps it could be altered somewhat to bring the reader around to the idea that such devices could have potential in MN testing but have not yet been tried.

Line 244: error message remains

Lines 254, 257: CCC should be spelled out or the abbreviation defined

Line 274: “Reference source not found” is embedded in the table description, and Eurofins is not completely spelled out

Line 280-281: contains “(Error! Reference source not found.”

Lines 291-302: issue with properly referring to and interpreting Figures 4A and 4B remain. Should “Figure 3A” at line 291 refer to Figure 4A? Is this intended to be figure legend or text interpretation of figures or a combination? Why are Figure 4A and Figure 4B listed without explanation at lines 299 and 302?

Line 308: change “and” to “which”? add “in” after “challenging to do”

Line 313: delete “and”

Line 325: this is the first that Rs is used to denote rho in the text; be consistent with terminology

Line 334: plasma is spelled incorrectly

There are assorted other minor issues with the discussion (eg. words compressed together) that I did not detail that could be resolved with further proofreading.

7. PLOS authors have the option to publish the peer review history of their article (what does this mean?). If published, this will include your full peer review and any attached files.

Reviewer #1: No

---

## [Author Response · Author response to Decision Letter 1]

28 Oct 2022

Responses to the Reviewer’s comments

Firstly, the project team would like to thank the Reviewer of this manuscript most sincerely for taking the time and effort to review our redrafting of the original manuscript. We were too hasty in trying to send back the edited document and therefore apologize for any time wasted by the Reviewer in light of a clean draft in being mistakenly labelled as in Track Changes. We have read and acted upon the Reviewer’s comments and our responses to them are below.

General: I am glad the authors found comments to be useful, and the responses and updated manuscript demonstrates that comments were thoughtfully considered. I did not see a track-changed version of the paper--it seemed like the original paper with updated figures and then a clean version of the updated paper was appended in the pdf I received--so comments below are based on what seemed to be the clean copy of the revision. Also, some of the issues that were pointed out in my original comments remain, so I hope that it was not a penultimate version that was inadvertently uploaded that I reviewed.

Response: We are very sorry for this mix up. We are not sure what happened, but we did go back and see that most of the original edits were acted upon but of course some were not and so again we apologize for any time wasted. In the newly submitted document in Track Changes, it is easier to see where all of these original edits were made in addition to our newer edits. The “Error” text issue is solved. We have now gone over the original review and added the comments from this secondary one and I believe that we have addressed everything noted by the Reviewer in both of their reviews. We now present a new document in Track Changes, with the hope that it will be much easier for the Reviewer to read our revisions in the text where they requested. 

1. Minor points—

Line 45: delete “in” 

Response: We have deleted “in utero” and left “fetus”. See Ln 41

2. Lines 97-99: sentence starting with “A variety of simple devices….” seems out of the blue; perhaps it could be altered somewhat to bring the reader around to the idea that such devices could have potential in MN testing but have not yet been tried.

Response: This sentence has been significantly modified to give greater clarity for the reader. Lns – 96 - 99.

“To support this, we propose that devices developed to passively fractionate plasma from whole blood in austere settings, primarily for viral load testing of people living with HIV [19,22], could be repurposed to include serum ferritin measurement and each of the other assays included on the MN 7-plex immunoarray.”

3. Line 244: error message remains

Response: This is no longer there. The entire document has also been checked for “Error”

4. Lines 254, 257: CCC should be spelled out or the abbreviation defined 

Response: We have defined the abbreviation in the text where first noted on Ln 221. We also added the full name in the first table legend to help the reader, Ln 254.

5. Real line 290 Line 274: “Reference source not found” is embedded in the table description, and Eurofins is not completely spelled out. 

Response: We have amended this on Lns 273 to read “* Eurofins data are excluded from sTfR results.”

6. Line 280-281: contains “(Error! Reference source not found.”

Response: This was resolved, we took out all the internal linking to table references, etc. to remove risk of seeing the field code errors again.

7. Lines 291-302: issue with properly referring to and interpreting Figures 4A and 4B remain. Should “Figure 3A” at line 291 refer to Figure 4A? Is this intended to be figure legend or text interpretation of figures or a combination? 

Response: We have added more text to provide greater clarity to the figure legends for Figs 4A and 4B. We have also deleted the reference to Fig 3A in the fig.4.A legend.

8. Why are Figure 4A and Figure 4B listed without explanation at lines 299 and 302?

Response: These were simply placeholders for where the figures were to be placed. We have deleted both of these with the assumption that the production editor will place each figure below its respective figurehead.

9. Line 308: change “and” to “which”? add “in” after “challenging to do”

Response: Done, see Ln 302.

10. Line 313: delete “and”

Response: Done, see Ln 312.

11. Line 325: this is the first that Rs is used to denote rho in the text; be consistent with terminology

Response: We have amended the text accordingly upon the first instance of Spearman’s correlation coefficient and have denoted either this and/or rho as Rs thereafter, in both the text and also in the figurehead for table 3 and in Table 3. 

12. Line 334: plasma is spelled incorrectly

Response: Corrected.

13. There are assorted other minor issues with the discussion (e.g., words compressed together) that I did not detail that could be resolved with further proofreading.

14. Response: We apologize for a poor review of the Discussion before sharing back with the Reviewer. We have had the document entire proofread in an attempt to eliminate any small typographic or spelling errors in the discussion and the remainder of the draft. A further example of this is where Q-plex was sometimes used to describe the assay we used, specifically this is the MN 7-plex assay and so it has been changed throughout. All are in track changes and so very clear to the Reviewer and the Editor.

---

## [Editor Report · Decision Letter 2]

4 Nov 2022

Multiplexed micronutrient, inflammation, and malarial antigenemia assessment using a plasma fractionation device

PONE-D-21-34846R2

Dear Dr. Boyle

We’re pleased to inform you that your manuscript has been judged scientifically suitable for publication and will be formally accepted for publication once it meets all outstanding technical requirements.

Kind regards,

José Luiz Fernandes Vieira

Academic Editor

PLOS ONE

Additional Editor Comments (optional):

All the suggestions of the reviewers were inserted in the manuscript

best regards

josé luiz vieira

---

## [Editor Report · Acceptance letter]

10 Nov 2022

PONE-D-21-34846R2 

Multiplexed micronutrient, inflammation, and malarial antigenemia assessment using a plasma fractionation device 

Dear Dr. Boyle:

I'm pleased to inform you that your manuscript has been deemed suitable for publication in PLOS ONE. Congratulations! Your manuscript is now with our production department. 

Kind regards, 

on behalf of

Dr. José Luiz Fernandes Vieira 

Academic Editor

PLOS ONE